# Bridging Human Vision and Deep Perception with a Saccade-Fixation ROI Prior for Medical Image Segmentation

## Abstract

Automatic medical image segmentation converts subjective visual interpretation into objective, pixel-level quantitative indicators with high precision and repeatability, providing essential morphological evidence for early disease detection and surgical planning. However, current segmentation networks universally follow an "equal-pixel" paradigm: every spatial location consumes the same amount of parameters regardless of its semantic saliency. Consequently, a large portion of computational resources are expended on lesion-free regions, leading to unnecessary GPU and memory overhead, and increasing the risk of overlooking tiny pathological areas. Human vision solves this problem through an active saccade-fixation strategy by first performing a rapid, low-resolution saccade to localize suspicious regions, then applying high-resolution fixation only where necessary. Inspired by this mechanism, we propose SaccadeFixationNet (SF-Net), a medical image segmentation framework that integrates biologically motivated gaze behaviors into an end-to-end trainable U-shaped architecture. SF-Net consists of a Saccade–Fixation Encoder (SFE) that combines global saccadic scanning with fixation-driven feature refinement, a Fixation Connectivity Module (FCM) that generates a Gaze ROI Map by modeling inter-fixation relations, and a Gaze-MoE Decoder (GMD) that adaptively routes fixation-relevant tokens to high-capacity experts while assigning peripheral regions to lightweight experts. This design enables ROI-guided selective computation, closely mimicking the allocation of neural resources in human vision. Extensive experiments on four heterogeneous medical datasets demonstrate that our model achieves significant performance gains and substantially outperforms baselines.

## 1 Introduction

Medical image segmentation is the cornerstone of precision medicine; by delineating organs and lesions at the pixel level it provides quantitative morphological evidence for surgical planning, therapy evaluation and early screening, and its accuracy directly affects subsequent dose calculation, surgical navigation and prognosis assessment, making it an indispensable core component of personalized healthcare. In recent years, medical image segmentation has achieved remarkable progress driven by deep learning. The CNN family effectively captures local textures and context through encoder-decoder skip connections, while Vision Transformer and its variants model long-range dependencies with self-attention, further improving global-structure understanding (Isensee et al., 2021).

However, these models universally follow an "equal-pixel" paradigm, applying homogeneous computation to every pixel or token, causing FLOPs to be wasted on regions without obvious pathology and exhibiting expensive quadratic complexity on ultra-high-resolution 3-D volumes (Shaker et al., 2024). Moreover, existing attention mechanisms usually make the "where to look" decision in a single forward pass, lacking the iterative refinement characteristic of human vision and easily missing tiny, low-contrast or diffuse lesions (Borji, 2024; Wang et al., 2025).

Human vision overcomes these bottlenecks through an active "saccade-fixation" cycle: a saccade quickly browses the entire scene at low resolution, while a fixation performs high-resolution scrutiny only within the fovea on locations of high uncertainty (Wloka et al., 2018). This cycle is uncertainty-

Figure 1: Analogy between human vision and SaccadeFixationNet. (a) Human vision uses saccades to rapidly scan a scene, fixations to focus on salient regions, and integration to recognize objects. (b) SF-Net mimics this cycle: the Saccade–Fixation Encoder captures global sweeps and local details, the Fixation Connectivity Module generates a Gaze ROI Map, and the Gaze-MoE Decoder allocates expert capacity to accurately reconstruct lesion regions.

driven—regions with large prediction error or low confidence trigger additional fixations—thus achieving accurate scene parsing with minimal samples (Samonds et al., 2018). Eye-tracking studies have shown that radiologists are more prone to false positives when their attention is overly diverted during X-ray reading (Good et al., 1990). Experts, in contrast, exhibit fewer fixations, shorter fixation durations, smaller saccadic amplitudes, and more efficient scan paths, enabling them to focus on lesion regions more quickly (Bertram et al., 2016). Therefore, it is crucial to rationally incorporate a "saccade-fixation" paradigm into the design of deep-model architectures for automatic and accurate medical image segmentation.

Inspired by the human saccade–fixation mechanism, we propose SaccadeFixationNet (SF-Net), which embeds the biological cycle of rapid scanning, focused fixation, and selective resource allocation into a U-shaped framework. As illustrated in Figure 1, SF-Net draws an analogy between human vision and computational design: (a) human perception alternates between coarse saccadic sweeps and fine fixations to recognize objects, while (b) SF-Net instantiates this cycle through dedicated modules. Specifically: (1) the Saccade–Fixation Encoder (SFE) mimics human vision by combining DINOv3-based (Siméoni et al., 2025) saccadic scanning with convolutional and Tok-KAN-based Li et al. (2025) fixation encoding, capturing both global semantic priors and fine structural details in a single forward pass; (2) the Fixation Connectivity Module (FCM) models inter-fixation relations to generate a Gaze ROI Map (G-Map), providing a structured prior that highlights regions most likely to be fixated and clinically relevant; (3) the Gaze-MoE Decoder (GMD) allocates heterogeneous expert capacity according to the G-Map: high-capacity experts (KAN-Expert, Hybrid-Expert) process fixation-relevant tokens, while lightweight experts handle peripheral tokens, enabling adaptive computation that mirrors selective neural resource allocation in human vision. The contributions of this paper are summarized as follows:

- To the best of our knowledge, this is the first work to formalize the human saccade–fixation mechanism into a U-shaped framework, with a Saccade–Fixation Encoder (SFE) that combines DINOv3-based scanning with convolutional and Tok-KAN-based fixation encoding.

- We introduce a Fixation Connectivity Module (FCM) that generates a Gaze ROI Map, and a Gaze-MoE Decoder (GMD) that allocates high-capacity experts to fixation regions and lightweight experts to peripheral regions, enabling ROI-guided selective computation without extra inference cost.

- We validate SF-Net on four heterogeneous 2D and 3D medical benchmarks, where it consistently outperforms state-of-the-art CNN-, Transformer-, Mamba-, and KAN-based models in accuracy and provides a new paradigm for high-precision, low-energy medical AI.

## 2 RELATED WORK

### 2.1 THE "EQUAL-PIXEL" PARADIGM IN MEDICAL IMAGE SEGMENTATION

U-Net (Ronneberger et al., 2015) and its derivatives (Attention U-Net (Oktay et al., 2018), U-Net++ (Zhou et al., 2018)) achieve robust results across modalities via encoder-decoder skip connections, yet they still adhere to the "equal-pixel" paradigm: every pixel shares the same number of convolutional kernels during forward-backward passes, causing $> 90\%$ of FLOPs to be expended on lesion-free regions. Vision Transformer variants (Swin-UNETR (Hatamizadeh et al., 2021), SegRes-Net (Myronenko, 2018)) incorporate global self-attention but treat all tokens equally; their quadratic complexity with image size hampers real-time inference of high-resolution 3-D volumes. Channel- or spatial-attention modules such as CBAM (Woo et al., 2018) and SE-Net (Hu et al., 2018) only re-weight features after a uniform backbone and do not prune redundant operations. Consequently, these methods exhibit low recall for small lesions in breast ultrasound and colorectal polyp datasets, confirming the inherent deficiency of "equal computation" in medically sparse-saliency scenes.

### 2.2 "NON-EQUAL-PIXEL" STRATEGIES AND THE ACCURACY-VS-FLOPS TRADE-OFF

Two research lines have been explored to mitigate computational redundancy. (1) Static lightweight designs: DeepMedic (Kamnitsas et al., 2016) employs multi-scale 3-D separable convolutions, reducing parameters by 5× but sacrificing 1.5–2.0 Dice points; HarDNet-MSEG (Huang et al., 2021) achieves 86 FPS at 0.9 mean Dice, yet lags behind heavy networks by 3–4 pp on glandular boundaries. (2) Dynamic inference / early-exit schemes: PointRend (Kirillov et al., 2020) iteratively samples MLPs on low-confidence pixels for re-segmentation, computing only 10 % of the region and boosting boundary IoU by 1.8 on Cityscapes; after porting to polyp data, overall Dice improves by 0.7, yet tiny-polyp recall shows no significant gain. SparseR-CNN (Sun et al., 2021) replaces dense anchors with 100 learnable proposals, cutting 35 % FLOPs on COCO; fine-tuned on a 2-D ultrasound breast dataset, IoU rises by 0.9 while recall for lesions $<5$ mm drops by 2.4 %.

These results show that merely trimming computation often compromises clinically critical metrics and still falls short of "seeing all while seeing well." In contrast, our SF-Net retains the overall budget but concentrates compute on the most uncertain regions via a learnable saccade-fixation cycle, simultaneously improving accuracy and efficiency.

### 2.3 MIXTURE-OF-EXPERTS (MOE) FRAMEWORK AND ITS POTENTIAL TO BREAK THE EQUAL-PIXEL PARADIGM

Recently, sparsely activated MoE has offered a "large-params-small-compute" alternative. Noisy Top-K gating (Shazeer et al., 2017) first reduced a 137 B-parameter network's inference cost to that of a 1 B dense model, validating sparse routing. GShard (Lepikhin et al., 2020) and Switch Transformer (Fedus et al., 2022) replace every other FFN with an MoE layer, training 600 B–1.6 T-parameter models while activating only 10–20 % experts, establishing the "sparse-is-efficient" paradigm. In vision, V-MoE (Riquelme et al., 2021) and Soft MoE (Puigcerver et al., 2023) introduce token-level routing to image classification, maintaining $> 90\%$ ImageNet Top-1 accuracy with 40 % fewer FLOPs.

In medical imaging, MoE's sparsity and specialization directly address the equal-pixel bottleneck: Background tokens are handled by a lightweight shared path, cutting GPU memory and latency significantly (Fedus et al., 2022). Modality- or organ-specific experts can be optimized independently, alleviating the "one-kernel-fits-all" contrast problem. The gating network can recall experts on demand according to uncertainty, realizing a "scan first, scrutinize later" second look and reducing false negatives from single-shot saliency (Puigcerver et al., 2023). Nevertheless, medical MoE still faces routing collapse, expert homogenization, and small-sample expert forgetting. By incorporating an uncertainty-driven fixation reward, SF-Net further lowers expert activation while maintaining high recall for tiny lesions, offering a scalable and interpretable sparse-expert route away from the equal-pixel paradigm.

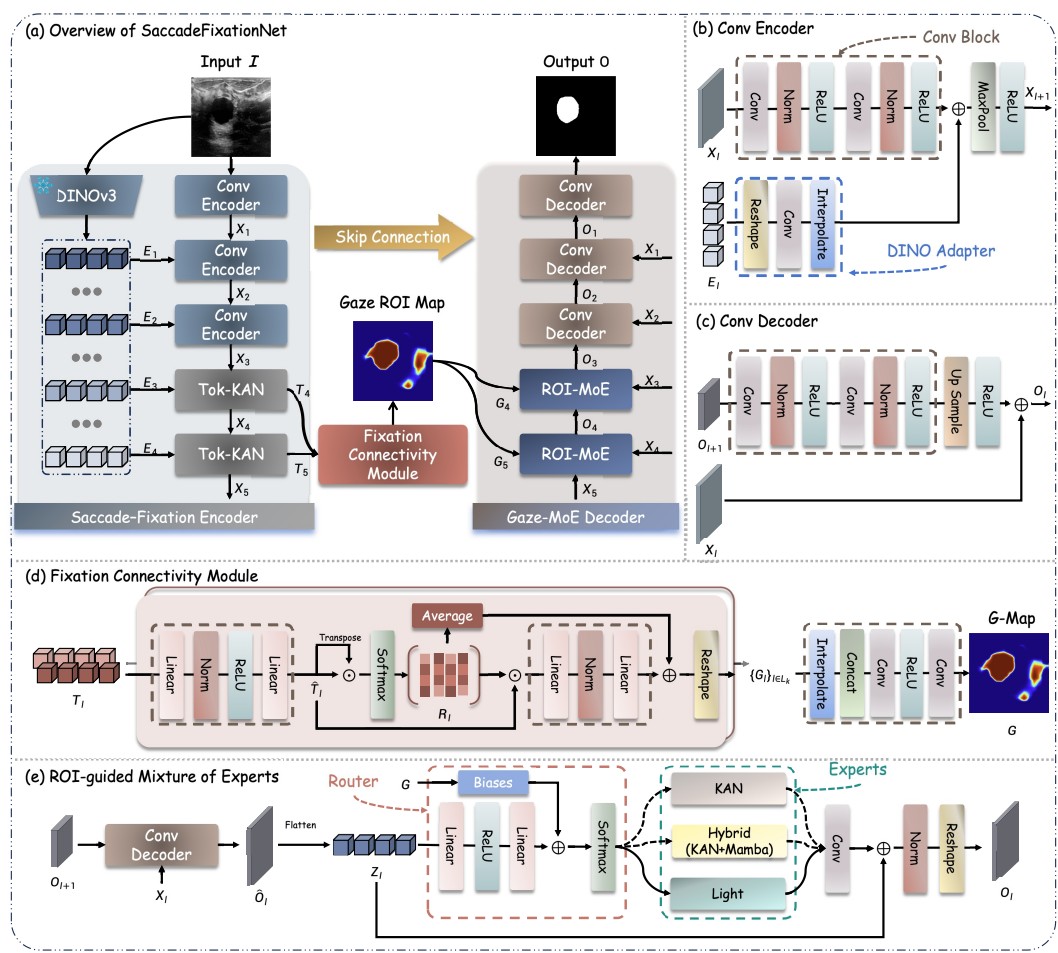

Figure 2: Architecture of SaccadeFixationNet (SF-Net). (a) Overall framework with Saccade–Fixation Encoder, Fixation Connectivity Module, and Gaze-MoE Decoder. (b) Conv Encoder with DINO adapter for token–spatial fusion. (c) Conv Decoder with progressive upsampling and skip connections. (d) Fixation Connectivity Module generating the G-Map from multi-level token relations. (e) ROI-guided Mixture of Experts with KAN, Hybrid (KAN–Mamba), and Light experts for adaptive decoding.

## 3 METHOD

### 3.1 OVERVIEW

We propose SaccadeFixationNet (SF-Net), a novel segmentation framework inspired by the human saccade–fixation mechanism. As illustrated in Figure 2, SF-Net consists of three major components: (1) Saccade–Fixation Encoder (SFE) that jointly captures saccade-level scanning and fixation-level refinement, (2) a Fixation Connectivity Module (FCM) that produces a Gaze ROI Map (G-Map) by modeling relations among fixation points, and (3) a Gaze-MoE Decoder (GMD) that allocates expert capacity adaptively according to the G-Map. Together, these components form a computational analogue of human vision: rapid saccades establish a broad perceptual prior, fixations refine salient details, and downstream neural pathways allocate resources preferentially to fixated regions.

### 3.2 SACCADE–FIXATION ENCODER (SFE)

The encoder is designed as a dual pathway that mirrors the complementary functions of saccade and fixation in human vision. The saccade path employs a DINOv3 Vision Transformer (Siméoni

et al., 2025) backbone to rapidly scan the input image and produce saccade features analogous to low-resolution visual sweeps. The fixation path consists of stacked convolutional encoders and Tok-KAN blocks (Li et al., 2025), which generate fixation features by attending to structural details during sustained viewing.

We adopt a pretrained DINOv3 as the saccade path and freeze all its parameters during training. Given an input image $I \in \mathbb{R}^{H \times W \times C}$, the backbone divides $I$ into $N = (H/p) \times (W/p)$ patch tokens and processes them through Transformer blocks to obtain $d$-dimensional token representations. To capture information across the visual sweep, we collect intermediate features from a subset of blocks

$$E_i = \text{DINO}_{M_i}(I), \quad E_i \in \mathbb{R}^{N \times d}, \quad i = 1, ..., D \tag{1}$$

where $M_i$ denotes the index of the selected Transformer block and $D$ is the number of extracted layers. These tokens serve as broad saccadic sweeps, providing semantic priors that roughly localize important structures.

The fixation path is defined as an encoder with $L$ stages, analogous to sustained fixations. The first $L_c$ stages are convolutional blocks (Conv+BN+ReLU+Pooling), which progressively downsample the input and capture structural details. The remaining stages adopt Tok-KAN blocks to model richer nonlinear dependencies. We set $X_0 = I$, and for each subsequent stage $\ell \in \{1, \dots, L\}$ the fixation representation $X_\ell \in \mathbb{R}^{H_\ell \times W_\ell \times C_\ell}$ is updated as

$$X_\ell = \begin{cases} \text{Pool}(\text{Conv}(X_{\ell-1})), & \ell \leq L_c, \\ \text{Tok-KAN}(X_{\ell-1}), & \ell > L_c, \end{cases} \tag{2}$$

To integrate the two pathways, each saccade token set $E_i$ is projected into a spatial feature map that can be aligned with fixation features by a DINO Adapter. This Adapter is achieved by a transformation

$$\phi_\ell : \mathbb{R}^{N \times d} \to \mathbb{R}^{H_\ell \times W_\ell \times C_\ell},$$

which consists of reshaping tokens into the corresponding spatial shape, applying a convolution layer for channel adaptation, and interpolating to the resolution of the fixation stage.

Since the fixation path has $L$ stages while the saccade path provides $D$ outputs with $D < L$, alignment is performed only for the deeper fixation stages. Specifically, the $i$-th saccade output is aligned to fixation stage $L - D + i$:

$$X_j \leftarrow X_j + \phi_i(E_i), \quad j = L - D + i, \ i = 1, \dots, D. \tag{3}$$

This SFE ensures that saccade features, which act as broad semantic sweeps, are progressively injected into deeper fixation stages that carry higher-level structural detail. Compared to conventional encoder–decoder designs where features are concatenated across scales, our additive integration provides a lightweight yet effective way to merge saccadic priors with fixation-driven representations, closely mimicking how human vision refines rapid sweeps into sustained fixations.

### 3.3 FIXATION CONNECTIVITY MODULE (FCM)

In human vision, fixations are not isolated events but exhibit structured connectivity, where attended points reinforce one another to form coherent gaze patterns. Inspired by this principle, we design the fixation connectivity module (FCM) to transform fixation tokens from the encoder into a structured prior, denoted as the Gaze ROI Map (G-Map).

We take the token outputs of selected Tok-KAN stages $L_k$. Each $T_\ell \in \mathbb{R}^{N_\ell \times C_\ell}, \ell \in L_k$ is first projected to enhance representational capacity and normalized to compute a relation matrix

$$\hat{T}_\ell = P(T_\ell), \quad R_\ell = \text{softmax}\left(\frac{\hat{T}_\ell \hat{T}_\ell^\top}{\tau}\right), \tag{4}$$

where $P(\cdot)$ is a learnable projection and $\tau$ a temperature parameter. $R_\ell \in \mathbb{R}^{N_\ell \times N_\ell}$ quantifies fixation connectivity by measuring token-to-token similarity.

From $R_\ell$, fixation importance is derived by combining the average attention weight of each token and an aggregated relational score. Formally, the fixation score is

$$s(i) = \frac{1}{N_\ell} \sum_{j=1}^{N_\ell} R_{\ell,ij} + \sigma\big(\Gamma(R_\ell \hat{T}_\ell)\big), \tag{5}$$

where $\Gamma(\cdot)$ is a lightweight aggregation network and $\sigma(\cdot)$ the sigmoid function. Reshaping $\{s(i)\}_{i=1}^{N_\ell}$ into $(H_\ell, W_\ell)$ yields a stage-level G-Map $G_\ell$.

To exploit fixation connectivity across scales, FCM generates fixation maps $\{G_\ell\}_{\ell \in L_k}$ and resizes them to a common resolution. These maps are concatenated and fused by a shallow convolutional head,

$$G = \sigma\Big(\text{Conv}\big(\text{Concat}\{G_\ell\}_{\ell \in L_k}\big)\Big). \tag{6}$$

The resulting G-Map $G$ serves as a biologically inspired prior, highlighting spatial regions with high fixation probability. It is subsequently used to guide decoding and expert allocation, ensuring that computational resources are focused on fixation-relevant regions.

## 3.4 GAZE-MOE DECODER (GMD)

The decoder progressively reconstructs the segmentation mask by alternating convolutional upsampling blocks and ROI-guided mixture-of-experts (MoE). At each stage, the feature map is first upsampled and processed by a convolutional decoder block, fused with the corresponding skip connection, and then selectively refined by a token-level MoE guided by the G-Map.

Formally, let $O_\ell$ be the fused feature map at decoding stage $\ell$. It is first processed by a convolutional decoder block and bilinearly upsampled:

$$\tilde{O}_\ell = U(\text{Conv}(O_{\ell+1})) + X_\ell, \tag{7}$$

where $U(\cdot)$ denotes bilinear upsampling and $X_\ell$ is the skip feature from the encoder. The resulting feature $\tilde{O}_\ell \in \mathbb{R}^{C_\ell \times H_\ell \times W_\ell}$ is then flattened into tokens $Z_\ell \in \mathbb{R}^{N_\ell \times C_\ell}$ with $N_\ell = H_\ell W_\ell$. Given the fixation prior $G_\ell$, the MoE update is defined as

$$w_\ell = \text{softmax}\Big(\tfrac{Q(Z_\ell)+\Delta(G_\ell)}{\tau}\Big), \tag{8}$$

$$\hat{Z}_\ell = Z_\ell + \sum_{e=1}^{E} w_{\ell,e} \odot \text{Expert}_e(Z_\ell), \tag{9}$$

$$O_\ell = \text{Reshape}(\hat{Z}_\ell), \tag{10}$$

where $Q(\cdot)$ is a linear projection mapping tokens to the routing space, $\Delta(G_\ell)$ provides ROI-dependent biases. We instantiate three heterogeneous experts: (i) a spline expert based on KAN (Liu et al., 2024b) blocks for nonlinear modeling, (ii) a hybrid expert combining Mamba (Gu & Dao, 2023) and KAN to capture sequential and structural dependencies, and (iii) a lightweight expert with linear and depthwise convolution for efficient background processing. Thus each stage consists of convolutional upsampling followed by ROI-guided expert refinement.

At later stages, when the resolution approaches the input size, only convolutional decoder blocks are used without MoE to refine boundary details. Finally, the output is resized to the original resolution $(H, W)$ and projected to segmentation logits:

$$\hat{Y} = \text{Conv}(O_1), \quad \hat{Y} \in \mathbb{R}^{H \times W \times C_{out}}. \tag{11}$$

This design ensures that convolutional decoding provides stable upsampling and skip fusion, while the ROI-guided MoE selectively enhances semantic features at intermediate scales. In this way, fixation-relevant regions are assigned to high-capacity experts, while peripheral regions are handled by lightweight experts, mimicking the resource allocation mechanism of human vision.

The training objective of SF-Net consists of two parts. First, we adopt a standard segmentation (cross-entropy) loss to supervise the final prediction $\hat{Y}$ against the ground-truth mask $Y$. Second, to regularize the fixation prior, we introduce ROI loss, which constrains the G-Map ($G$) to align with the foreground regions. Since $Y$ may contain multiple classes, we define a foreground mask function $F(Y)$ that maps all non-background pixels to 1 and background pixels to 0. The regularization is then written as

$$\mathcal{L} = \text{CE}(Y, \hat{Y}) + \lambda\,\text{BCE}(F(Y), G) \tag{12}$$

where $\text{BCE}(\cdot)$ denotes binary cross-entropy and $\lambda$ is a balancing weight.

Table 1: Comparison with state-of-the-art segmentation models on three heterogeneous medical scenarios. The average results with standard deviation over three random runs are reported.

| Methods | BUSI (Al-Dhabyani et al., 2020) | | GlaS (Valanarasu et al., 2021) | | CVC (Bernal et al., 2015) | |
|---|---|---|---|---|---|---|
| | IoU↑ | F1↑ | IoU↑ | F1↑ | IoU↑ | F1↑ |
| U-Net (Ronneberger et al., 2015) | 57.22±4.74 | 71.91±3.54 | 86.66±0.91 | 92.79±0.56 | 83.79±0.77 | 91.06±0.47 |
| Att-Unet (Oktay et al., 2018) | 55.18±3.61 | 70.22±2.88 | 86.84±1.19 | 92.89±0.65 | 84.52±0.51 | 91.46±0.25 |
| U-Net++ (Zhou et al., 2018) | 57.41±4.77 | 72.11±3.90 | 87.07±0.76 | 92.96±0.44 | 84.61±1.47 | 91.53±0.88 |
| U-NeXt (Valanarasu & Patel, 2022) | 59.06±1.03 | 73.08±1.32 | 84.51±0.37 | 91.55±0.23 | 74.83±0.24 | 85.36±0.17 |
| Rolling-UNet (Liu et al., 2024a) | 61.00±0.64 | 74.67±1.24 | 86.42±0.96 | 92.63±0.62 | 82.87±1.42 | 90.48±0.83 |
| U-Mamba (Ma et al., 2024) | 61.81±3.24 | 75.55±3.01 | 87.01±0.39 | 93.02±0.24 | 84.79±0.58 | 91.63±0.39 |
| U-KAN (Li et al., 2025) | 63.38±2.83 | 76.40±2.90 | 87.64±0.32 | 93.37±0.16 | 85.05±0.53 | 91.88±0.29 |
| SF-Net | **68.41±4.02** | **80.52±3.04** | **88.68±0.54** | **93.98±0.28** | **86.02±0.44** | **92.41±0.26** |

## 4 EXPERIMENTS AND RESULTS

### 4.1 DATASETS AND IMPLEMENTATION DETAILS

We evaluate SF-Net on four heterogeneous medical segmentation datasets. The 2D benchmarks include **BUSI** (Al-Dhabyani et al., 2020) with 647 breast ultrasound images covering normal, benign, and malignant cases (resized to $256 \times 256$), **GlaS** (Valanarasu et al., 2021) with 165 annotated histology images (resized to $512 \times 512$), and **CVC-ClinicDB** (Bernal et al., 2015) with 612 colonoscopy frames extracted from 31 video sequences (resized to $256 \times 256$). For 3D volumetric segmentation, we adopt **BraTS2025**, which is evaluated under two configurations: (i) *Pre-only*, including 1,251 pre-treatment training cases with four MRI modalities (T1, T1ce, T2, FLAIR) and annotations for three tumor subregions—enhancing tumor (ET), tumor core (TC), and whole tumor (WT); and (ii) *Pre+Post*, extending to both pre- and post-treatment volumes, totaling 2,818 training cases and covering four tumor-related structures—ET, TC, WT, and resection cavity (RC).

For 2D datasets, we follow the implementation and evaluation setting in U-KAN (Li et al., 2025). The dataset was randomly split into 80% training and 20% validation subsets. Results are reported over three random runs. For BraTS2025, following the experimental protocol of SegMamba (Xing et al., 2024), we split those datasets into training/validation/testing sets using a 70%/10%/20% ratio. We adopt a 3D crop size of $(64 \times 64 \times 64)$ and a batch size of 2. Training runs for 1,000 epochs with data augmentations including brightness, gamma, rotation, scaling, mirror, and elastic deformation

Unless otherwise specified, SF-Net is configured with $L_c = 3$ convolutional blocks in the fixation path ($L = 5$), and $D = 4$ saccade outputs from DINOv3. In the Gaze-MoE decoder, we employ three heterogeneous experts (KAN, Hybrid Mamba–KAN, and Light), with Top-1 routing as the default setting. For 2D tasks, standard 2D convolutions are used throughout the network, while for 3D volumetric tasks, all convolutional layers are replaced by their 3D counterparts without altering the overall architecture.

### 4.2 RESULTS ON 2D BENCHMARKS

We compare SF-Net with representative segmentation models, including U-Net (Ronneberger et al., 2015), Attention U-Net (Oktay et al., 2018), U-Net++ (Zhou et al., 2018), U-NeXt (Valanarasu & Patel, 2022), Rolling-UNet (Liu et al., 2024a), U-Mamba (Ma et al., 2024), and U-KAN (Li et al., 2025). Quantitative results on BUSI, GlaS, and CVC are shown in Table 1.

Across all datasets, SF-Net consistently achieves superior performance over existing methods. On BUSI, SF-Net obtains 68.41% IoU and 80.52% F1, clearly surpassing U-KAN, demonstrating the benefit of integrating fixation priors into breast ultrasound segmentation. On GlaS, SF-Net achieves 88.68% IoU and 93.98% F1, improving over U-KAN by +1.0 IoU and +0.6 F1, which highlights its robustness on histology data. On CVC-ClinicDB, SF-Net reaches 86.02% IoU and 92.41% F1, again outperforming U-KAN by nearly +1 IoU and +0.5 F1, showing the ability of our gaze-inspired design to capture small and irregular polyp structures.

Overall, SF-Net surpasses both classical baselines, as well as more advanced KAN- and Mamba-based designs. The most notable improvements are observed on BUSI and CVC, where fixation priors and ROI-guided decoding are particularly beneficial for segmenting small, heterogeneous lesions. Qualitative visualizations further demonstrate that SF-Net provides sharper boundaries and fewer false positives compared to existing methods.

Table 2: Comparison of segmentation performance across different models on BraTS 2025 (Pre+Post). Metrics include DSC (higher is better) and HD95 (lower is better) for RC, ET, TC, and WT.

| Models | RC | | ET | | TC | | WT | |
|---|---|---|---|---|---|---|---|---|
| | DSC ↑ | HD95 ↓ | DSC ↑ | HD95 ↓ | DSC ↑ | HD95 ↓ | DSC ↑ | HD95 ↓ |
| UNETR(Hatamizadeh et al., 2022) | 51.05 | 22.44 | 72.33 | 10.01 | 71.75 | 11.17 | 85.95 | 8.21 |
| SwinUNETR (Hatamizadeh et al., 2021) | 67.27 | 14.62 | 74.99 | 8.93 | 75.81 | 9.30 | 89.42 | 5.63 |
| SegResNet (Myronenko, 2018) | 74.91 | 10.09 | 74.86 | 8.42 | 75.70 | 8.68 | 89.43 | 5.36 |
| SegMamba (Xing et al., 2024) | 76.16 | 10.25 | 77.20 | 7.50 | 78.20 | 7.79 | 90.02 | 4.75 |
| SF-Net | **84.41** | **5.28** | **80.29** | **5.99** | **81.14** | **6.34** | **91.63** | **3.69** |

Table 3: Comparison of segmentation performance across different models on the BraTS 2025 (Pre). Metrics include Dice (higher is better) and HD95 (lower is better) for ET, TC, and WT.

| Models | ET | | TC | | WT | |
|---|---|---|---|---|---|---|
| | Dice ↑ | HD95 ↓ | Dice ↑ | HD95 ↓ | Dice ↑ | HD95 ↓ |
| UNETR (Hatamizadeh et al., 2022) | 83.69 | 5.85 | 89.36 | 5.43 | 91.93 | 5.64 |
| SwinUNETR (Hatamizadeh et al., 2021) | 85.52 | 4.48 | 91.73 | 4.02 | 93.11 | 5.02 |
| SegResNet (Myronenko, 2018) | 86.39 | 4.16 | 91.30 | 3.91 | 93.18 | 4.29 |
| SegMamba (Xing et al., 2024) | 86.69 | 4.53 | 91.90 | 4.27 | 93.32 | 4.67 |
| SF-Net | **87.52** | **3.79** | **91.79** | **3.27** | **93.83** | **3.46** |

## 4.3 RESULTS ON 3D BRAIN TUMOR BENCHMARK

To assess the generalizability of SF-Net beyond 2D images, we conduct experiments on the BraTS 2025 3D brain tumor segmentation benchmark. We report Dice score (DSC) and Hausdorff distance (HD95) for tumor subregions. Results compared with UNETR (Hatamizadeh et al., 2022), Swin-UNETR (Hatamizadeh et al., 2021), SegResNet (Myronenko, 2018), and SegMamba (Xing et al., 2024) are summarized in Tables 2 and 3.

On the BraTS 2025 Pre+Post setting (Table 2), SF-Net substantially outperforms all baselines across all subregions. For the challenging RC class, SF-Net achieves 84.41% DSC and 5.28 mm HD95, outperforming SegMamba by +8.3 DSC and reducing HD95 by 5 mm. Consistent gains are also observed for ET, TC, and WT, with DSC improvements of 2–3 points and HD95 reduced by 2–4 mm compared to the strongest baselines. On the BraTS 2025 Pre-only setting (Table 3), which contains only pre-treatment scans, SF-Net still delivers the best results. It achieves 87.52% DSC on ET, 91.79% on TC, and 93.83% on WT, while also obtaining the lowest HD95 values across all categories. These results demonstrate that SF-Net generalizes well across imaging protocols and remains robust even under modality constraints.

Overall, the BraTS 2025 results confirm that SF-Net is not limited to 2D tasks but extends effectively to 3D volumetric segmentation. The fixation connectivity prior and ROI-guided MoE decoder provide consistent improvements in both accuracy and boundary precision, particularly for complex, multi-component tumor structures.

## 4.4 ABLATION STUDY

To assess the contribution of each component in SF-Net, we perform ablation experiments on the 2D datasets, with results summarized in Table 4. Removing the saccade path (w/o DINO) leads to a noticeable performance drop across all benchmarks confirming that global semantic sweeps from DINO provide essential priors. Similarly, removing the fixation connectivity module (w/o FCM) consistently degrades results, especially on CVC, highlighting the importance of modeling fixation relations for accurate localization of small and irregular structures.

We also investigate different routing strategies in the Gaze-MoE decoder. While sparse Top-$k$ routing (w/ Top2 or w/ Top3) achieves competitive performance, it generally underperforms the default Top-1 routing. For example, on BUSI, Top-3 routing yields 66.78% IoU compared to 68.41% with Top-1 routing. This shows that assigning each token to its most relevant expert (Top-1) is more stable and better exploits fixation priors than distributing tokens across multiple experts.

Table 4: Ablation study on the effect of the saccade path (DINO), fixation connectivity (FCM) and Top-$k$ routing.

| Methods | BUSI (Al-Dhabyani et al., 2020) | | GlaS (Valanarasu et al., 2021) | | CVC (Bernal et al., 2015) | |
|---|---|---|---|---|---|---|
| | IoU↑ | F1↑ | IoU↑ | F1↑ | IoU↑ | F1↑ |
| SF-Net | **68.41±4.02** | **80.52±3.04** | **88.68±0.54** | **93.98±0.28** | **86.02±0.44** | **92.41±0.26** |
| SF-Net w/o DINO | 67.23±5.04 | 79.53±3.96 | 86.51±0.85 | 92.72±0.46 | 85.21±0.53 | 91.92±0.35 |
| SF-Net w/o FCM | 67.42±5.50 | 79.62±4.31 | 85.92±0.78 | 92.39±0.44 | 84.43±1.61 | 91.44±0.95 |
| SF-Net w/ Top2 | 68.13±5.01 | 80.06±3.98 | 88.63±0.67 | 93.94±0.36 | 85.40±2.21 | 92.04±1.33 |
| SF-Net w/ Top3 | 66.78±4.83 | 79.26±3.79 | 88.64±0.71 | 93.95±0.38 | 85.83±0.86 | 92.32±0.53 |

Overall, SF-Net achieves the best performance among all variants. These results verify that the saccade path, fixation connectivity, and ROI-guided MoE routing are complementary, and their integration is necessary to achieve consistent improvements across heterogeneous datasets.

## 4.5 VISUALIZATION

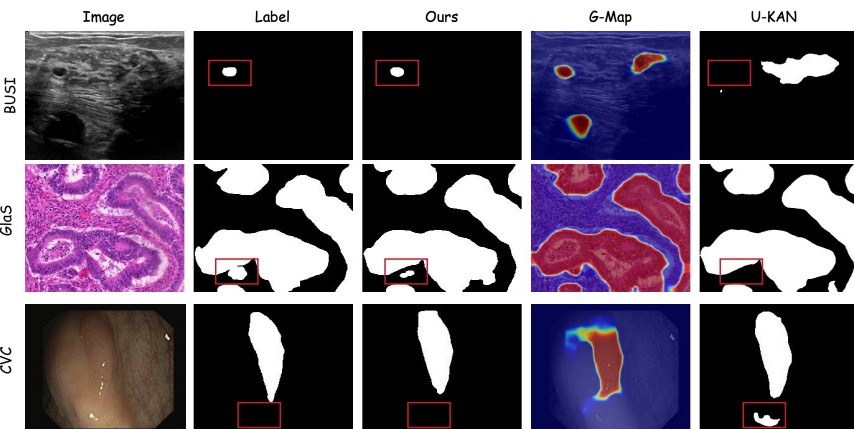

Figure 3: Qualitative comparison of segmentation results on BUSI, GlaS, and CVC datasets. From left to right: input image, ground-truth label, SF-Net prediction, G-Map, and U-KAN prediction.

To further illustrate the effectiveness of SF-Net, we provide qualitative comparisons on representative samples from 2D datasets, as shown in Figure 3. Columns correspond to the input image, ground-truth label, our segmentation results, the generated G-Map, and predictions from U-KAN.

Across three datasets, SF-Net produces more accurate and robust delineations than U-KAN, particularly in challenging regions highlighted by red boxes. On BUSI, our method captures small tumor regions with higher fidelity, while U-KAN tends to over-segment. On GlaS, SF-Net better preserves fine gland boundaries and detects small isolated structures that U-KAN misses. On CVC, our model eliminates false positives and provides tighter polyp contours. The G-Map visualizations further demonstrate that the fixation connectivity prior effectively highlights clinically relevant regions, guiding the decoder to focus computational resources where errors are most likely to occur.

## 5 CONCLUSION

In this paper, we presented SF-Net, a biologically inspired framework that embeds the saccade–fixation mechanism into a U-shaped architecture. SF-Net integrates a Saccade–Fixation Encoder, a Fixation Connectivity Module generating a Gaze ROI Map, and a Gaze-MoE Decoder for ROI-guided expert allocation. Experiments on four heterogeneous 2D and 3D medical benchmarks show that SF-Net consistently surpasses state-of-the-art CNN-, Transformer-, Mamba-, and KAN-based models, achieving more accurate lesion segmentation with improved efficiency. This work demonstrates the potential of gaze-inspired designs for advancing precise, efficient, and interpretable medical image analysis.

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

## A   APPENDIX

You may include other additional sections here.

