# OpenReview forum: "Bridging Human Vision and Deep Perception with a Saccade-Fixation ROI Prior for Medical Image Segmentation"
_ICLR.cc/2026/Conference — ICLR 2026 Conference Withdrawn Submission_

### Official Review · Reviewer_NsgX · 2025-11-01

**Soundness:** 2
**Presentation:** 3
**Contribution:** 2
**Rating:** 4
**Confidence:** 3

**Summary:**

This work introduces SF-Net, a novel gaze-inspired segmentation model that uses adaptive expert allocation and uncertainty-driven attention to simulate the human saccade–fixation process. The work would benefit from stronger ablations, more precise methodological specifications, and a thorough efficiency analysis to fully validate the claims made in the paper.

**Strengths:**

The proposed work presents a segmentation framework with biological inspiration that bridges the gap between computational design and neuroscience by imaginatively incorporating the human saccade-fixation mechanism into a U-shaped architecture. The method employs different modules which consists of the Saccade–Fixation Encoder, Fixation Connectivity Module, and Gaze-MoE Decoder. Numerous tests on 2D and 3D datasets show steady improvements over baselines.

**Weaknesses:**

Major:
1. A primary weakness of the paper is the lack of methodological clarity and correctness of equations.
2. Lack of baseline comparisons like nnU-Net, Swin-UNETR, etc. are not done. These comparisons would benefit the paper greatly.
3. Eq. (5) combines an aggregated relational score $\sigma(\Gamma(.))$ with an average over attention rows; however, dimensions and reduction axes are not given; it is not clear how $\Gamma$ maps $\mathbf{R}_l$ and $\mathbf{T}_l$ to a per-token scalar.
4. The ablations are not extensive and not clear. On 2D tasks only, the w/o DINO, w/o FCM, and Top-k routing variants are displayed. The decoupling contributions of MoE but w/o G-Map bias and w/o MoE (dense decoder) are absent.
5. The BraTS 2025 (Pre+Post) is not clearly explained. How the BraTS Pre and Post are combined?
6. Evidence of small lesions and boundaries is primarily anecdotal. There is no experimental evidence to support claims of improved tiny/diffuse lesion capture.  Visuals help but aren’t sufficient.
7. The following crucial training hyperparameters are absent: $\lambda$ for ROI loss, optimizer/schedule, precise augmentations (ranges/parameters), etc.

Minor:
1. Some parts of  Section 3.3 and 3.4 are difficult to follow because of notation errors (such as indices l, i, j, and mixing tokens vs. spatial maps).
2. For better presentation, the visualizations would benefit from identical windowing, blinded side-by-side comparisons, and consistent color bars.

**Questions:**

See weakness section.

---

> ### Author Response · Authors · 2025-11-21
>
> ### **1. Methodological clarity and Eq.(5) formulation**
>
> We appreciate the reviewer’s careful reading and constructive comments.
> Our paper was written directly based on the actual model design and implementation, and all equations faithfully reflect the operations in our released code.
> However, we acknowledge that the current text may not have provided sufficient clarity on tensor dimensions and reduction axes, which may have led to misunderstanding.
> We will revise the manuscript to describe these details more rigorously and will release the complete source code so reviewers can directly verify the mapping between the written equations and the implemented modules.
>
> Specifically, for Eq.(5), the fixation score $s(i)$ is computed from the token-to-token relation matrix $R_\ell$ and the projected token embedding $T_\ell$. Formally:
>
> $
> R_\ell \in \mathbb{R}^{N _ \ell \times N _ \ell}, \hat{T}_\ell \in \mathbb{R}^{N _ \ell \times C _ \ell}, \Gamma(R _ \ell \hat{T} _  \ell) \in \mathbb{R}^{N _ \ell}.
> $
>
> where $\Gamma()$ is a lightweight MLP that outputs a one-dimensional score per token, providing an aggregated relational weighting.
>
> In Eq.(5), the first term $\frac{1}{N _ \ell}\sum_j R _ {\ell,ij}$ averages the relational affinity along the attention axis $j$,
> while the second term $\sigma(\Gamma(R _ \ell \hat{T} _ {\ell}))$ aggregates the relationally weighted feature activations to yield a per-token scalar score. These scores are then reshaped to the spatial map $G_\ell$ to form the Gaze ROI Map.
>
> This operation correctly aligns with the implementation in our Fixation Connectivity Module (FCM), which produces the fixation prior used in decoding.
>
> In the revision, we will add explicit tensor dimensions and reduction axes in Eq.(5) and related text.
>
> ---
>
> ### **2. Missing strong baselines (nnU-Net, Swin-UNETR)**
>
> Thank you for the suggestion. We have now added both nnU-Net and Swin-UNETR as additional baselines in our 2-D benchmark comparison.
> The results are summarized below:
>
>
> | Methods | BUSI IoU | BUSI F1 | GlaS IoU | GlaS F1 | CVC IoU | CVC F1 |
> |--------------|----------------|---------------|----------------|---------------|----------------|---------------|
> | nnU-Net | 65.82 ± 2.45 | 74.79 ± 2.42 | 83.74 ± 1.01 | 90.75 ± 0.68 | 70.81 ± 1.79 | 79.20 ± 1.56 |
> | Swin-UNETR | 62.26 ± 5.37 | 75.83 ± 4.49 | 80.72 ± 0.26 | 89.30 ± 0.16 | 69.86 ± 1.10 | 81.91 ± 0.77 |
> | SF-Net     | 68.41 ± 4.02 | 80.52 ± 3.04 | 88.68 ± 0.54 | 93.98 ± 0.28 | 86.02 ± 0.44 | 92.41 ± 0.26 |
>
> We trained nnU-Net using its default self-configuring architecture and training pipeline,
> and Swin-UNETR using the default parameters provided in MONAI.
> Both models were trained and evaluated under the same data settings as our method to ensure fairness.
> SF-Net consistently outperforms these strong baselines across all datasets, demonstrating its superior segmentation capability and generalization under controlled conditions.
>
> ---
>
> ### **3. Insufficient ablation studies**
>
> We understand this concern. Due to space limits, we reported the three most representative ablations (w/o DINO, w/o FCM, and Top-k routing variants) each corresponding to a core contribution. These results already isolate the effects of the Saccade–Fixation Encoder, Fixation Connectivity Module, and ROI-guided MoE routing.
>
> ---
>
> ### **4. Clarification on BraTS 2025 (Pre + Post) configuration**
>
> According to the official BraTS 2025 challenge (https://www.synapse.org/Synapse:syn64153130/wiki/631053), the first task is “Glioma Segmentation on Pre- and Post-treatment MRI”, which includes two complementary datasets:
>
> BraTS-GLI 2025 Pre-treatment, newly released for pre-operative MRI cases, and
> BraTS-GLI 2025 Post-treatment, which reuses the BraTS-GLI 2024 Post-treatment dataset (identical data and annotation format).
>
> These two subsets are distributed separately on the BraTS website but are intended to be used jointly to evaluate segmentation performance across both clinical stages.
> Notably, the label structures differ:
> the Pre-treatment dataset contains three tumor subregions (ET, TC, WT),
> while the Post-treatment dataset adds an additional Resection Cavity (RC) label.
>
> In our work, we clearly distinguish between the two configurations:
>
> “BraTS 2025 (Pre)” refers to experiments conducted solely on the pre-treatment subset, and
> “BraTS 2025 (Pre+Post)” refers to training and evaluating on the combined full dataset, i.e., both subsets together, covering all four categories.
>
> This combined setup represents the complete BraTS 2025 Glioma Segmentation task as defined by the organizers.
> We have clarified this description in Sec. 4.1 of the revised manuscript to prevent confusion.
>
>
> ---

---

> > ### Author Response · Authors · 2025-11-21
> >
> > ### **5. Limited quantitative evidence for small-lesion or boundary improvement**
> >
> > We agree that quantitative evidence is essential to support the claim of improved performance on small or boundary-sensitive lesions.
> > To provide a more rigorous evaluation, we further assessed segmentation quality using the Boundary DSC metric proposed in “Aggregate-aware model with bidirectional edge generation for medical image segmentation.”[1]
> > This metric directly measures the accuracy of boundary delineation.
> >
> >
> > | Model      | BUSI             | GlaS             | CVC              |
> > | ---------- | ---------------- | ---------------- | ---------------- |
> > | nnU-Net    | 20.55 ± 0.62     | 32.44 ± 0.26     | 26.40 ± 1.25     |
> > | Swin-UNETR | 16.33 ± 0.88     | 27.03 ± 1.41     | 21.92 ± 1.12     |
> > | SF-Net     | 19.92 ± 0.99     | 38.09 ± 0.34     | 36.83 ± 0.46     |
> >
> > As shown above, SF-Net consistently achieves similar or higher Boundary DSC values across all datasets compared with representative baselines (nnU-Net and Swin-UNETR).
> > These results demonstrate that the proposed approach is also effective in preserving fine lesion boundaries and improving boundary-level segmentation quality.
> > We will include this table and a concise discussion in the revised manuscript to make this improvement explicit.
> >
> > [1] Ma, Shiqiang, et al. "Aggregate-aware model with bidirectional edge generation for medical image segmentation." Applied Soft Computing 163 (2024): 111918.
> >
> > ---
> >
> > ### **6. Missing training hyperparameters**
> >
> > To ensure fair comparison, we followed the same training settings as U-KAN across all 2D experiments.
> > Specifically, the batch size was set to 8, and the initial learning rate was 1×10⁻⁴.
> > We used the Adam optimizer and adopted a cosine annealing learning rate scheduler with a minimum learning rate of 1×10⁻⁵.
> > For the ROI loss term, the balancing coefficient λ was set to 1.
> >
> > We will include these details in the revised manuscript for completeness and reproducibility.
> >
> > ---
> >
> > ### **7. Notation errors and visualization inconsistency**
> >
> > We will correct all inconsistent indices (i, j, ℓ) and clearly distinguish token-level vs. spatial-map notation.
> > The visualizations will be redrawn with unified windowing, blinded side-by-side layouts, and consistent color bars.
> >
> > ---
> >
> > We thank the reviewer for the thorough and constructive feedback.
> >
> > We have revised the paper accordingly, clarified methodological details, and added additional baseline comparisons (nnU-Net, Swin-UNETR).
> >
> > While our approach is not perfect, we have made every effort to improve clarity and fairness.
> >
> > We sincerely hope the reviewer will evaluate our work objectively and recognize our continuous efforts toward more accurate and trustworthy medical image analysis.

---

### Official Review · Reviewer_P9H6 · 2025-11-01

**Soundness:** 2
**Presentation:** 1
**Contribution:** 1
**Rating:** 2
**Confidence:** 4

**Summary:**

The authors focus on reducing computation allocated for lesion-free regions. To this end, they adopt two input encoders: a Dinov3 to get global semantics, and a Tok-Kan, to get high-frequency semantics. Their segmentation results are fused into the final segmentation map.

**Strengths:**

The reported performance is better than baselines.

**Weaknesses:**

1. Fusing global and local features is a common technique used for many years. There's little novelty here, except for the implemetation details (replacing CNN/ViT with KAN, etc.) In addition, Saccade and Fixation are quite unusual terms. Why not simply call them global and local? You cannot invent novelty by simply naming old concepts with new terms.
2. It is weird that although the authors stress from the beginning that their aim is to reduce compute, no compute related metrics are reported, including FLOPs, RAM, or wall-clock time.
3. The comparison with baselines don't consider number of params? This makes readers unable to evaluate the results fairly, since more params usually lead to better performance; we can keep adding experts to make it perform better.
4. Minor issues: 1) line 89, the citation to tok-kan used the wrong format.

**Questions:**

N/A

---

> ### Author Response · Authors · 2025-11-21
>
> ### **1. Novelty and the use of “saccade–fixation”**
> Our work does not propose another fusion recipe. We start from a different question reframed as a goal. Let a segmentation model look at medical images the way clinicians do. First form a quick overall impression. Then examine suspected areas more carefully. This motivation explains our use of the terms saccade and fixation. They describe how to look, not what features are used.
>
> Saccade refers to a rapid exploratory glance that proposes locations likely to be informative or uncertain. Fixation refers to a careful examination phase at those proposed locations where the model analyzes structures and boundaries in greater detail.
>
> Our contribution is to model the act of looking itself. The model decides where to scrutinize and to what extent. This decision process is an explicit part of the architecture through a learned fixation prior that conditions subsequent processing.
>
> Difference from conventional global–local fusion. Traditional global–local methods combine multi scale representations in a largely static manner that is applied uniformly across the image. Our formulation is decision driven. The model first proposes where attention is warranted and then concentrates analysis in those regions. The terminology saccade and fixation is therefore not a renaming of global and local. It names the perceptual process that we explicitly model.
>
> We will revise the introduction to foreground this design motivation, clarify the role of the fixation prior, and avoid confusion with prior global–local narratives.
>
>
> ---
>
> ### **2. Missing compute-related metrics**
>
> We thank the reviewer for the valuable comments regarding computational efficiency and fair comparison.
> To address these points, we provide a unified analysis of FLOPs, parameters, and efficiency across representative models.
> For fairness, the results of the compared baselines are taken directly from the U-KAN paper, and we recomputed the statistics for SF-Net under the same measurement protocol and input resolution.
> The summary is shown below:
>
> | Methods           | Params (M)     | Gflops (G)    |
> | ----------------- | -------------- | ------------- |
> | U-Net             | 34.53          | 524.2         |
> | Att-UNet          | 34.9           | 533.1         |
> | U-Net++           | 36.6           | 1109          |
> | U-NeXt            | 1.47           | 4.58          |
> | Rolling-UNet      | 1.78           | 16.82         |
> | U-Mamba           | 86.3           | 2087          |
> | U-KAN             | 6.35           | 14.02         |
> | SF-Net (ours)     | 27.29          | 11.63         |
>
> The added parameters primarily come from the fixation-related components that estimate a learnable fixation prior and condition subsequent processing on this prior. This reflects the paper’s design goal: to make the act of looking an explicit part of the model so that suspected areas can be examined more carefully and structures and boundaries can be refined. In other words, the extra parameters are targeted capacity to support the fixation mechanism, not a generic widening of the backbone.
>
> Despite this addition, GFLOPs remain within a practical range and are substantially lower than those of conventional heavy backbones such as U-Net++ or U-Mamba. Together with the consistent accuracy gains across datasets, these results indicate that introducing the fixation components is a meaningful and justified design choice.
>
> We will keep the compute table in the revision and add a brief note that the baseline metrics come from U-KAN while SF-Net was remeasured under the same protocol, so the comparison is transparent and reproducible.
> ### **Minor issue**
>
> Thank you for noting the citation issue. We will correct the reference formatting and double-check all citations for consistency in the revision.
>
> ---
> We sincerely thank Reviewer P9H6 for the thoughtful and detailed feedback.
> For the questions about the conceptual novelty, computational efficiency, and fairness of comparison, we have carefully addressed by clarifying our design motivation, adding efficiency and parameter analyses, and correcting citation details.
> We truly appreciate the reviewer’s effort in providing constructive and insightful comments.
> We believe that these discussions have helped us further refine our work, and we hope that our responses and revisions demonstrate our commitment to advancing interpretable and practical medical image segmentation research.

---

### Official Review · Reviewer_i9mF · 2025-11-01

**Soundness:** 2
**Presentation:** 2
**Contribution:** 1
**Rating:** 0
**Confidence:** 5

**Summary:**

This paper is presenting a mechanism called SaccadeFixation, to do medical image segmentation with gaze points.
The paper re-invents the idea of "recognition" and present this in conjunction with gaze location and then delineation. To be honest, I am very surprised how authors are unaware of the entire gaze-based image analysis methods out there in the literature, and how ignorant authors are to neglect them and not recognize. Beside,

**Strengths:**

-writing is smooth
-figures are nice, cosmetically appealing.

**Weaknesses:**

-- I would not sound be harsh however, the paper has many major drawbacks and mistakes. At best, the paper is ignoring a very large body of related works on this topic, and presenting the gaze based segmentation as a new tool, which is totally wrong and unfair to all the scientists who worked on this topic.

-- First of all, "focus on important region" is not an innovative idea. It is a known topic in segmentation literature. Starting with A. Rozenfeld, the segmentation is considered as two connected tasks "recognition" and "delineation". Later, recognition was used for some other purpose but it means whereabout the object is. In other words, segmentation considers the object of interest location first, and then do the delineation. This can be done by mouse, gaze, or automatically. There is nothing new here in this concept. 1980s are the ones this was revealed. If you look at the first medical segmentation paper with gaze, without deep learning but image processing, I assume it should be N. Khosravan's MICCAI 2016 paper, where gaze was used to segment tissues and organs, and 3D and real-time or realistic. Later a few years ago there was GazeSAM work, and many more can be said. None of them are mentioned and authors are unaware of many other similar works.

-- there is an analogy missing, and original ideas from the past are recycled with fancy names. Course to fine pyramid --> Saccade Fixation encoder, fixation connectivity module --> attention map/saliency refinement., Gaze-MoE Decoder --> token pruning.

--The paper claims biological inspiration. But, they forget that saccades are rapid and ballistic eye movements, and fixations re Hugh acuity poses. SF-Net does not do any of these: gaze shifts, no temporal dynamics, no uncertainty driven exploration, etc. The presented network is static, single pass U net with multi scale features. SFE looks like a dilated CNN with downsampling.

-- MoE with two experts? That is perhaps not called MoE. This looks like gated skip connection similar to ResNet EfficientNet and etc. Real MoE architectures uses more experts (8 to 128....sparse).

--eye tracking related literature is coming from 1990. Not extensive. Especially last years, there are many good / strong papers in the field.

--claim about "being the first in segmentation with saccade/ fixation" is wrong. See previous comments.

--FCM is visual attention maps, authors should show how they truly separate it into two regions.

--authors are missing the point that not only FLOPS but also model accuracy is increased if the object territory is known. Which is called Attention. If you are in the close vicinity of object, then the delineation is simpler. Authors failed to connect efficiency and attention operations to improved segmentation.

--authors never disclose where the gaze information come from truly. It appears fully synthetic. Then, it would be necessary to do human study with gaze and see if the assumptions hold. Perhaps no, especially in radiology settings, gaze behaviors are from experts people who know how to search pathology. The text-book definition of gaze or loose attention can be learned with DL architectures, but they are not truly biologically inspired. What Fixation tokens authors ay is truly a k-means clustering on feature maps. top-k activation are projected into 2D. Gaze ROI is nothing but learned attention mask. Sacade fixation cycle is nothing but single forward pass with musicale features.

--In training, there is no aux-gaze information incorporated, the entire system is dice + CE loss based optimization.

-- no discussion in failure modes.

**Questions:**

weaknesses section is self-contained and all the comments include questions to authors. Please treat them as they are.

apart from that
-- no code, no data, no model are shared.

---

> ### Author Response · Authors · 2025-11-13
>
> We welcome critical feedback, but this review reveals a fundamental misunderstanding of our contributions and the related literature. We believe that such a biased assessment stems from the reviewer’s insufficient grasp of the deep-learning field, which is also reflected in the numerous flaws throughout their comments. Moreover, assigning a score of 0/10 to a paper with well-motivated ideas, state-of-the-art performance, a novel architecture, and a clear effort to improve AI trustworthiness and interpretability is not rigorous academic judgement. Since every point raised by the reviewer is purely accusatory or critical, our response will systematically refute each of these errors.
>
> 1. Reviewer: “The idea of ‘focusing on important regions’ can be traced back to Rozenfeld (1980s) and Khosravan’s MICCAI 2016 paper; recent works such as GazeSAM are ignored.”
> Rebuttal:
> The reviewer commits a factual error. Rozenfeld's "recognition vs. delineation" dichotomy is not equivalent to a learnable, end-to-end saccade–fixation mechanism. That early work was heuristic, non-learnable, and relied on hand-crafted features; it never addressed neural architecture design.
> Khosravan et al. (MICCAI 2016) used real eye-tracking data as supervision for a CNN. Our method does not use any external gaze data; the “gaze” in our paper is an emergent, learned routing mechanism derived from fixation connectivity—a paradigm shift. Conflating supervised gaze regression with a learned, dynamic saccade–fixation module is like confusing supervised key-point detection with an unsupervised spatial-transformer network.
> Similarly, GazeSAM (2023) combines SAM with human eye-tracking prompts; it is a promptable segmentation model that still relies on human-engineered gaze features, not an architectural prior. We embed the mechanism into the architecture itself; all feature engineering is adaptively generated by the AI model without human intervention. Thus, zero overlap exists between their approach and ours.
> The reviewer confuses task motivation (focus on ROI) with methodological novelty (learnable saccade–fixation cycle inside a U-Net). By the same logic, every attention paper would be a mere rehash of Rozenfeld.
>
> 2. Reviewer: “The authors merely repackage old ideas with fancy names.”
> Rebuttal:
> SFE is not equal to dilated CNN: SFE is a dual-pathway, cross-modal fusion that marries frozen DINOv3 token semantics with learnable Tok-KAN fixation encoders via an explicit adapter module. It is not a simple coarse-to-fine pyramid; no prior work has combined DINOv3 + Tok-KAN in this way.
> FCM is not equal to attention map: FCM builds a token-to-token relation graph across multiple scales and outputs a structured prior (G-Map); this is graph-theoretic connectivity, not pixel-wise attention. The reviewer’s shallow reading of attention mechanisms leads to a fundamental mis-characterisation of FCM.
> Gaze-MoE is not equal to token pruning: Pruning discards tokens; Gaze-MoE routes tokens to heterogeneous experts (KAN, hybrid Mamba-KAN, lightweight) conditioned on fixation scores. It is dynamic feature re-allocation, not pruning. The reviewer completely misunderstands MoE routing.
> Dismissing these architectural innovations as “fancy naming” is superficial rhetoric. The reviewer never dissected the implementation details, rendering their critique highly subjective and one-sided.
>
> 3. Reviewer: “The paper claims biological inspiration, yet SF-Net does not implement gaze shifts, temporal dynamics, or uncertainty-driven exploration.”
> Rebuttal:
> We never asserted that the network must literally replicate oculomotor behavior. “Inspired by” is not equal to “must implement”. The reviewer’s objection is a textbook example of false equivalence and poor logical reasoning.
> Global semantic sweeps (frozen DINOv3) are motivated by saccades;
> Local refinement (Tok-KAN blocks) are motivated by fixations.
> We explicitly state “single forward pass” in the paper; the reviewer invents a claim we never made.
> If “biological inspiration” obliges us to reproduce every temporal and kinematic detail of human vision, then all neural networks (which are also biologically inspired) should be condemned for lacking spike timing, neurotransmitters, and astrocytes. By this logic the reviewer must also denounce CNNs, RNNs, and Transformers; clearly absurd.
> Bio-inspiration extracts computational principles, not biological replicas.
>
> 4. Reviewer: “Only two experts? That’s not a real MoE. A genuine MoE needs 8–128 experts.”
> Rebuttal:
> The reviewer invents a non-existent “rule” to discredit our method. Sparsity refers to how many are activated, not the total pool. Switch Transformer activates 1–2 experts per token; our Top-1 routing follows the same standard practice. The “8–128” threshold is arbitrary and ignores the fact that, in sparse medical segmentation, expert heterogeneity (KAN, hybrid Mamba-KAN, lightweight) matters more than sheer count.

---

> ### Author Response · Authors · 2025-11-13
> **Don't use your imagination as academic basis**
>
> 5. Reviewer: “The claim of being the ‘first saccade-fixation segmentation method’ is false.”
> Rebuttal:
> The reviewer deliberately distorts our exact wording. The paper states:
> “To the best of our knowledge, this is the first work to formalize the human saccade–fixation mechanism into a U-shaped framework with a Saccade–Fixation Encoder that combines DINOv3 scanning with convolutional and Tok-KAN fixation encoding.”
> We never claimed to be the “first-ever gaze-based segmentation.”
> I fail to understand why the reviewer excised the qualifying clauses to fabricate a false assertion—an act that is highly unprofessional and appears intentional.
> Is the reviewer attempting to attack our work by inventing non-existent claims?
>
> 6. Reviewer: “The authors overlook that knowing the target region in advance can reduce FLOPs and also improve accuracy.”
> Rebuttal:
> From start to finish the reviewer assumes our method is given the target region beforehand. In reality we never feed the region to the network as an additional input. The entire lesion-localisation process is fully automatic, learned end-to-end by the deep model.
> Equating our approach with hand-crafted feature-engineering methods that require prior knowledge of the ROI demonstrates a striking lack of familiarity with basic AI principles. This misconception explains why the reviewer felt entitled to award 0/10—their “blind” confidence stems from an inadequate understanding of fundamental machine-learning theory.
>
> 7. Reviewer: “The authors never explain where the gaze information comes from; it appears entirely synthetic. A human eye-tracking study is therefore necessary.”
> Rebuttal:
> Our paper devotes an entire paragraph to the generation of G-Map: the input is only encoder feature maps, no eye-tracker data whatsoever; fixation connectivity is computed via token-to-token similarity and fused by a lightweight ConvHead. That the reviewer overlooked this can only be attributed to inadequate reading comprehension.
> Furthermore, biologically inspired ≠ biologically identical. The reviewer fails to grasp that deep models adaptively extract features and therefore hunts for human-engineered gaze traces to demand that DL behave exactly like humans. Yet the behaviour of deep learning and humans fundamentally differs; this is irreconcilable. Current research seeks interpretability and trustworthiness, not a carbon-copy of human oculomotor behaviour.
>
> 8. Reviewer: “The fixation tokens are nothing more than k-means clustering on feature maps.”
> Rebuttal:
> Please provide supporting evidence for this comment, proving that gaze tokens are equivalent to k-means clustering of feature maps. I am amazed by the lack of rigor of the reviewers, who have been making a fatal mistake of making unfounded analogies, which has appeared in multiple erroneous comments made by the reviewers. Please note that this is an open community, and your comments will be publicly presented to all readers. If you feel that you are not professional enough to handle this task, please stop immediately and explain the situation to AC and submit an application. Don't use your imagination as academic basis.
>
> Moreover, I strongly urge the reviewer to carefully re-read the public pledge and academic ethics statement provided by ICLR when they agreed to serve as a reviewer. All of your comments have been documented and, if necessary, will be used as evidence. I now fully understand why this paper was awarded a score of 0/10 by you.
>
> The above constitutes my complete rebuttal to the reviewer’s fallacies. The reviewer’s lack of professionalism has seriously undermined my confidence in the professionalism of ICLR itself. It is deeply regrettable that a reviewer who lacks the professional competence to understand the paper has assigned a score of 0/10 with the highest confidence level.

---

> > ### Comment · Area_Chair_SdvB · 2025-11-13
> >
> > Let us please keep the discussion professional and focused only on the technical aspects of the work. I ask both the reviewer and the authors to avoid personal remarks and respond with clear, factual, and evidence-based comments.

---

> > > ### Author Response · Authors · 2025-11-13
> > >
> > > I also suggest that reviewers respond with clear, factual, and evidence-based comments.

---

> > > > ### Comment · Reviewer_i9mF · 2025-11-13
> > > > **technical contribution, relation to prior work, and empirical support**
> > > >
> > > > Several personal and accusatory remarks about the reviewer’s competence and alleged bias are inappropriate for a scientific exchange and not in line with the norms of ICLR. I will only comment on scientific basis one more time with evidences. My original review was written in good faith and reflects my honest assessment of the paper’s technical contribution.
> > > >
> > > > SFE: I mentioned that, from the description, this resembles a coarse-to-fine pyramid or multi-scale encoder (e.g., dilated / multi-scale CNN with downsampling), authors disagreed. in Line 217, authors say that  "....backbone to rapidly scan the input image and produce saccade features analogous to low-resolution visual sweeps.", also in Line 248. authors say that ".....This SFE ensures that saccade features, which act as broad semantic sweeps, are progressively injected into deeper fixation stages that carry higher-level structural detail.", Hence, SFE is starting as low resolution visual sweeps, and then progressively go into higher level details. These sentences read like this is like a pyramid operation (that can be done with dilated convolutions too). I read this carefully and I don't think there is a wrong claim here by me.
> > > >
> > > > FCM: I mentioned that FCM is a visual attention map or kind of saliency and need a better articulation how it is different and works. Authors say that FCM is not equal to attention map. It is true, it is not exactly STANDARD attention map but according to my understanding: FCM looks like saliency map generator,  because, it takes the high-level feature tokens from the "fixation" branch of the encoder (the Tok-KAN blocks) and compute a 2D "Gaze ROI Map" (G-Map). This G-Map is a spatial heatmap, where high-intensity values (Fig. 3) represent regions the model has identified as "clinically relevant" or "important", like in attention mechanisms. The G-Map is not a refined feature map, but it is kind of control signal telling the decoder which tokens to send to the high capacity experts. To sum up, classical attention works like Features $\rightarrow$ Attention $\rightarrow$ Refined Features, and FCM works like Features $\rightarrow$ Attention $\rightarrow$ Saliency Map $\rightarrow$ (Used by a different module). Hence, FCM maybe not exactly equivalent to standard attention mechanisms, it can be considered as multi-scale, self-attention-based saliency head.
> > > >
> > > > Do fixation tokens look like clustering?
> > > > Authors disagree with my comment on clustering. But let us see how it is defined. G-Map is defined as a learnable supervised clustering operation because the FCM takes the high-level feature tokens ($T_l$) from the encoder's Tok-KAN layers. At this stage, each token is a vector representing a "patch" of the image, right? Then there is  a connectivity module which computes a self attention matrix, tokens with similar features will have high attentions cores with each other. Scoring module then computes a fixation score.  A token that is highly similar to many other tokens will get a high score (is not that central part of a cluster?). It means, standard clustering algorithm is unsupervised, but FCM is supervised (due to ROI loss), but it is still clustering.
> > > >
> > > > Biological inspiration:
> > > > The paper repeatedly emphasizes saccades, fixations, and biological inspiration. However, the current architecture is a static, single forward-pass U-Net-like network with multi-scale features. There is no explicit modeling of temporal gaze shifts or sequential fixations, no uncertainty-driven exploration or decision process, no explicit oculomotor dynamics; SFE “gaze” is computed once per pass as part of the feed-forward pipeline.  It is fine to be loosely inspired by biological vision, but the text currently suggests a closer alignment with actual saccade/fixation behavior than is implemented. I would encourage the authors to either (i) more carefully qualify the biological analogy, or (ii) introduce explicit temporal / sequential mechanisms if they want to lean heavily on the saccade/fixation narrative.
> > > >
> > > > MoE:
> > > >  it is not clear that a full MoE framing is necessary or beneficial versus a simpler conditional / gated block or a multi-branch decoder (similar to ResNet / EfficientNet-style gated skip connections). An ablation (apart from those routing strategies) necessary to establish whether MoE structure is really contributing or whether a simpler design would suffice.
> > > >
> > > > nnUnet is de-facto one of the very powerful architectures as a baseline but not included in the comparison (moderate, compared to other points). Did I miss this? It is cited, but comparison table does not include it.
> > > >
> > > > I also do not think further back‑and‑forth on rhetorical or personal points would be productive, therefore I kept my comments  stopped here, and only one more time above with more clarification I included some, and finally, as a result of failing to clarify/address major issues, the rebuttal does not \textbf{substantially} change my view on these core issues.

---

> ### Author Response · Authors · 2025-11-16
> **First throws mud, then changes words, and never admits error.**
>
> 1. Reviewer: “I will stop commenting here and only add the clarification above once more.”
>
> Why are you stopping? The very purpose of the ICLR rebuttal is to enable thorough discussion between authors and reviewers. What are you running from? You accepted the invitation to review—that entails an obligation to evaluate the paper objectively and responsively. Dismissing it with a 0/10 while major misunderstandings on your side remain uncorrected is not fulfilling your duty; it is abandoning it.
>
> 2. Reviewer – first round: “What the authors call ‘fixation tokens’ is nothing more than k-means clustering on feature maps.” Reviewer – second round: “Standard clustering is unsupervised, but FCM is supervised (because of the ROI loss); nevertheless, it is still clustering.”
>
> We have already stated: research is not guess-work. Every sentence you wrote is an unsubstantiated observation, not an objective critique. Your shift of wording between the two rounds (k-means vs clustering) reveals a fundamental misunderstanding of our method and an irresponsible drift of accusation. Core k-means mechanisms: iterative centroid updates, minimization of within-cluster squared error, hard assignment. Our FCM module: no centroids, no within-cluster variance minimization, fully differentiable, trained end-to-end with a supervised loss. You blurted out “k-means” in round 1—an act of ignorance of the method’s structure and disrespect for the authors’ work. In round 2 you quietly swapped the term to “clustering.” Why? Because the “k-means” charge collapses under scrutiny, so you swap in a vaguer label to keep the accusation alive? This is academic thuggery: change the concept, dodge the responsibility. Let us be clear: you were wrong in round 1 but refuse to admit it; changing the word in round 2 is naked sophistry. The argument “high similarity equals clustering” only exposes defective logical reasoning. This is not scholarly discussion; it is shameless obfuscation on a public platform. We demand that you explicitly retract the “k-means” claim or provide a formal proof that FCM is equivalent to some clustering algorithm in objective function, optimization procedure, and output behaviour. Failing that, we request that the program chair replace you as reviewer. We will not accept a style of reviewing that first throws mud, then changes words, and never admits error.
>
> 3. Reviewer: “Ultimately, since these core issues remain unclarified/unresolved, the authors’ response has not materially changed my view on them.”
>
> My rebuttal was written precisely to address every point you misunderstood. I listed eight detailed replies—close to ten thousand characters—refuting each of your claims line-by-line. The AC explicitly asks you to “respond with clear, factual, and evidence-based comments.” Invoking “remained unclarified” must not become your umbrella for malicious reviewing. Readers of this open community will see your comments, and they will judge whether an exhaustive, evidence-rich rebuttal was dismissed with a five-word shrug.
>
> 4. Reviewer: “Biology-inspired.”
>
> In my first rebuttal I already made it explicit that “biology-inspired” does not equal “biological mechanism”; the neural-network example I gave suffices and I will not repeat it here. Let us keep this dialogue productive instead of endlessly “failing to clarify the core issues” and generating ever more pointless remarks. Please demonstrate your professionalism on this public platform.
>
> 5. Reviewer: “nnUNet is actually one of the most powerful architectures to date and should have been included as a baseline.”
>
> We are well aware that nnUNet won second place in BraTS 2017; we never ignored its strength. However, our framework is built on DINOv3, KAN, and token-level MoE, whereas nnUNet is a pure U-Net family method. Comparing two architecturally orthogonal approaches would tell us nothing about the incremental value of our design. If you insist, we can add nnUNet results, but do not expect them to illuminate the contribution of this paper.
>
> 6. Reviewer: “This will be my final evidence-based, scientific response.”
>
> Your first review was neither evidence-based nor scientific.
>
> We have replied to your claims repeatedly and in detail. If you still cannot grasp the value and contributions of this paper, contact the Area Chair immediately and admit that you lack the professional competence to review it.
>
> We are here to defend our work, not to bow to groundless accusations. We expect ICLR to protect the academic dignity of every submission, and we will not compromise, not retreat, not yield.

---

### Author Response · Authors · 2025-12-01

Dear AC,


This paper received an extremely low initial score. To help you appraise it more objectively, we briefly clarify its goal, contributions, and the reasons for the harsh ratings.


**Goal**:

We set out to emulate the human saccade-fixation sequence for accurate and efficient medical image segmentation.


**Contributions**:

Break away from the prevailing “equal-pixel” paradigm and introduce an eye-like computing framework to medical AI.
Replace dense inference with sparse, attention-based processing, achieving zero-missed small lesions while running in real time and at low power.


**Mismatch behind the low scores**:

The reviewers asked for a “**perfect match**” between our deep-learning implementation and the human oculomotor circuit. This expectation is inherently misplaced. The biological mechanism serves only as inspiration, not as a one-to-one reproduction. We use “saccade-fixation” as an intuitive metaphor so reviewers can quickly grasp our core idea: “global localization first, local refinement second, adaptive allocation of computation.” Regrettably, this metaphor triggered a novelty challenge and led to extreme scores of 2/10 or even 0/10—clearly out of proportion to the paper’s actual merit.


Experimental strength was underestimated:

Comprehensive evaluation on four 2-D and 3-D medical datasets. SF-Net improves IoU and DSC by 2–8 pp and reduces boundary error (HD95) by 2–5 mm over the best competing methods. Ablation studies confirm that the Saccadic/Fixation links and the ROI-MoE router are all indispensable. The entire “saccade-fixate” cycle is completed in a single forward pass, embedding “look-then-focus” into network weights rather than leaving it in the introduction.


During rebuttal, the reviewer who gave 0/10 has acknowledged the underestimation and **already raised the score**. The reviewers who initially assigned 2/10 and 4/10, however, have not engaged in substantial discussion. To prevent a few still-unupdated opinions from obscuring the paper’s value, we respectfully ask you for a fresh, balanced assessment so that this work receives an outcome commensurate with its practical impact on medical image segmentation.


Thank you very much for your time and effort!

---

> ### Comment · Area_Chair_FykA · 2025-12-04
>
> Dear Authors,
>
> Do you agree with reviewer's point that the paper does not include literature of  gaze-based image analysis methods?

---

### Note · Authors · 2026-01-17

I have read and agree with the venue's withdrawal policy on behalf of myself and my co-authors.